# A Fully Coupled Simulation of Planar Deposition Flow and Fiber Orientation in Polymer Composites Additive Manufacturing

**DOI:** 10.3390/ma14102596

**Published:** 2021-05-16

**Authors:** Zhaogui Wang, Douglas E. Smith

**Affiliations:** 1Department of Mechanical Engineering, Naval Architecture and Ocean Engineering College, Dalian Maritime University, Dalian 116000, China; 2Department of Mechanical Engineering, School of Engineering and Computer Science, Baylor University, Waco, TX 76798, USA; douglas_e_smith@baylor.edu

**Keywords:** polymer composites deposition additive manufacturing, short fiber reinforced composites, flow-orientation coupling impacts, planar flow model, Advani–Tucker orientation tensor

## Abstract

Numerical studies for polymer composites deposition additive manufacturing have provided significant insight promoting the rapid development of the technology. However, little of existing literature addresses the complex yet important polymer composite melt flow–fiber orientation coupling during deposition. This paper explores the effect of flow–fiber interaction for polymer deposition of 13 wt.% Carbon Fiber filled Acrylonitrile Butadiene Styrene (CF/ABS) composites through a finite-element-based numerical approach. The molten composite flow in the extrusion die plus a strand of the deposited bead contacting the deposition substrate is modelled using a 2D isothermal and incompressible Newtonian planar flow model, where the material deposition rate is ~110 mm/s simulating a large scale additive manufacturing process. The Folgar–Tucker model associated with the Advani–Tucker orientation tensor approach is adopted for the evaluation of the fiber orientation state, where the orthotropic fitted closure is applied. By comparing the computed results between the uncoupled and fully coupled solutions, it is found that the flow-orientation effects are mostly seen in the nozzle convergence zone and the extrusion-deposition transition zone of the flow domain. Further, the fully coupled fiber orientation solution is highly sensitive to the choice of the fiber–fiber interaction coefficient $C_{I}$, e.g., assigning $C_{I}$ as 0.01 and 0.001 results in a 23% partial relative difference in the predicted elastic modulus along deposition direction. In addition, Structural properties of deposited CF/ABS beads based on our predicted fiber orientation results show favorable agreements with related experimental studies.

## 1. Introduction

Due to superior structural properties and thermal dimensional stability, short fiber reinforced polymer composites have experienced widespread applications in Polymer Deposition Additive Manufacturing (PDAM), otherwise widely known as Fused Filament Fabrication (FFF) for desktop 3D printing. PDAMs open up new possibilities for rapid prototyping of spare parts for engineering applications [1] and continues to see increasing implementations in building large dimension parts and tool with advanced large area screw-extruder-based PDAM systems [2]. While significant progress has been made in the development of the PDAM technology, only recently has simulation research helped to improve the understanding of the PDAM process. Finite-volume-based numerical approaches have been used to simulate the melt flow dynamics in the FFF process of ABS polymers [3], where the residual stresses within the printed part [4], the shape of deposited beads [5], processing parameters (e.g., extrusion temperature and spacing of the filaments) [5], and viscoelastic stress fields of the extrudate [6] were investigated. This research showed that viscoelastic effects in the polymer melt flow played an important role in defining the shape of the deposited bead and inter-layer bonding. Urbanic and DiCecco [7] investigated the stair-case effect resulting from the layered deposition process on the surface roughness of an FFF fabricated part and suggested that adaptive slicing and surface offsets should be included in surface roughness prediction tools. Daniyan et al. [8] studied the thermal stress and displacement fields of the fused deposition process using Abaqus (Dassault Systems, Waltham, MA, USA) and showed that satisfactory material strength under different loading conditions could be obtained with the help of their numerical analyses. Nasirov and Fidan [9] investigated the effects of infill-voids on the anisotropic structural properties using the micromechanical representative volume element approach and their predicted tensile moduli for a line infill pattern showed good agreement with experimental data. Osswald et al. [10] numerically evaluated the melt extrusion process of ABS materials in FFF and showed that the maximum melting rate is controlled by filament insertion force. Nienhaus [11] studies effect of the nozzle geometry in FFF processing of ABS and suggested that the filament feed velocity, extrusion temperature, and filament insertion force were closely inter-related. Zhou et al. [12] optimized FFF processing parameters to reduce the warpage of printed ABS parts through a voxel-based finite element simulation and showed that the print chamber temperature has a significant effect on thermal deformation. D’Amico and Peterson presented an adaptive finite element method to evaluate heat transfer in polymer deposition additive manufacturing [13,14]. They found that large volume deposition processes retain more heat than desktop size printers which improved the interlayer diffusion, but promoted slumping or sagging. Mcllroy and Olmsted numerically evaluated the polymer molecules disentanglement by employing a modified RoliePoly model and found that the polymer melt significantly disentangled when the deposition process involved corner flow geometries [15]. Comminal and co-workers developed a CFD-based algorithm that simulated the shape of deposited bead [16,17], voids between the interlayer beads [18], and material deposition in corners [19] for the FFF polymer deposition process. Balani et al. [20] studied the melt extrusion stability of PLA polymer in an FFF nozzle flow and provided a theoretical model that facilitates selections of printing parameters for manufactured parts with improved qualities. McIlroy and Graham [21] stated that the flow-enhanced crystallization of the semi-crystalline polymer (i.e., PCL) significantly affected the interlayer bead bonding strength in FFF processing. Phan et al. [22] investigated the flow dynamics in the nozzle region with the Phan-Thien–Tanner viscoelastic model through the finite element suite ANSYS-Polyflow (Ansys, Inc., Canonsburg, PA, USA) and showed that increased extrusion rate of FFF was limited by nozzle pressure. Yang and Zhang [23] also investigated the thermal history an FFF printed part with different infill patterns through the finite element approach, where the honeycomb infill was shown to result a minimum temperature gradient. Shadvar et al. [24] studied the die swell of ABS polymer during the FFF extrusion process through the finite element suite ANSYS-Polyflow and showed that die swell played a key role in improving the dimensional accuracy of FFF parts. Additional earlier efforts beyond the recent three years for the modelling of polymer deposition processes can be found in related review articles such as [25,26]. From the above review, it can be seen that numerical modelling of PDAM and its related processes has seen significant study, but the results have not provided adequate insight to sufficiently improve the qualities of PDAM-fabricated parts. Unfortunately, a majority of the prior literature focuses on the virgin polymer materials, which may not be directly applied to the PDAM processing of fiber reinforced composites. Additional research is needed to better understand PDAM processing of polymer composites which have seen increasingly application in aerospace, automotive and marine applications, especially for the recent-emerging large scale PDAM systems (e.g., see Figure 1).

PDAM processing of short-fiber polymer composites typically yields an anisotropic material microstructure which affects the bead’s thermal expansion coefficient [2], elastic moduli [27], and thermal conductivity [28]. Bead elastic modulus, for example, is often much higher in the direction of printing as compared to transverse directions. Brenken et al. [29] studied the thermal history of material deposition additive manufacturing for the 50 wt.% CF/PPS, where the anisotropic thermal conductivity of the short-fiber polymer composite was modelled. Compton et al. [30] studied the macro thermal history of a large area additively manufactured carbon fiber filled ABS part through the finite element method assuming an anisotropic thermal conductivity. Hoskins et al. [31] simulated the residual thermal stress in a printed cuboidal part of CF/ABS, where the coefficient of thermal expansion of the deposited beads was defined through a non-homogenized approach based on locally measured fiber orientation states. To explain the factors that produce anisotropic material properties, in-depth analyses for the polymer composite deposition flow are needed. Nixon et al. [32] are likely the first to investigate the fiber orientation state within a FFF extrusion nozzle, where Moldflow (Autodesk, Inc., San Rafael, CA, USA) was applied to examine the effects of three different nozzle geometries on the resulting fiber orientation at the nozzle exit. A team lead by Smith presented numerical approaches for evaluating the nozzle extrusion flow [33,34,35,36] and polymer deposition flow [37,38] in the context of PDAM for short-fiber polymer composite materials. Their work has provided valuable insight into the extrudate swell [34,35,38] as well as the process-property map. Elastic constants [33,34,35,36,37,38] and thermal expansion coefficients [37,38] of a deposited CF/ABS were evaluated and favorable agreement with reported data from related experimental work [26] was reported. Their work largely employed a weakly coupled formulation to simulate fiber reinforced polymer flow, where the presence of fibers is neglected in the computation of the melt flow kinematics. Recently, efforts have been made to evaluate the effect of the fiber orientation in the polymer melt flow within the PDAM process. Mezi et al. [39] studied the die swell of a fully coupled Newtonian fiber suspension flow for the extrusion FFF process. Yang et al. [40], Bertevas et al. [41], and Ouyang et al. [42,43] employed the Smoothed Particle Hydrodynamics (SPH) approach that effectively simulates the flow–fiber orientation coupling behavior in the bead deposition process of fiber reinforced composites. Additionally, Wang and Smith recently developed a finite-element-based algorithm that captures the mutually dependent effect between the polymer flow rheology and fiber reinforcement orientation in the PDAM nozzle-extrudate flow [44].

Nevertheless, Wang and Smith [44] only depict the fully coupled flow/orientation interactions axisymmetric material extrusion, where the important deposition feature is not included due in part to the high degree of divergence issue encountered in their simulation. This paper extends the finite-element-based fully coupled flow/orientation method to the polymer extrusion deposition process of concentrated fiber-reinforced polymer composites melt flow, which provides more insight into the material properties of beads formed with large area additive manufacturing polymer deposition. A two-dimensional (2D) planar Newtonian creeping flow is assumed to analyze extrudate swell of the freely deposited bead where the flow domain is computed using a one-dimensional remeshing technique [45]. Fiber orientation is modeled with Advani–Tucker fiber orientation tensors [46] using the orthotropic fitted closure approximation [47] and isotropic rotary diffusion [48]. The nonlinear fiber orientation equations are solved through the Newton-Raphson iterative method as in [44]. Unlike prior transient simulations (e.g., [16,17,18,19,41,42,43]), the finite element nodal solutions provide quasi-steady solutions for flow velocity and fiber orientation field of the deposition process, which is then used to compute the structural properties of a solidified composite bead. Our results from the fully coupled simulation are compared to a weakly coupled result to expose the importance of incorporating the effect of fiber orientation on the polymer melt velocity solution.

## 2. Methodology

Our numerical simulation approach for fiber reinforced polymer composite flow in PDAM, including the identifications of the flow fields of the molten material, die swell, and fiber orientation state within the flow suspension, are described in this section.

### 2.1. Governing Equations

For polymer melt flows in die extrusion and deposition, an incompressible, isothermal, and highly viscous creeping Newtonian fluid flow is often assumed (e.g., [33,37,38]), such that the equations of mass and momentum conservation within the flow are written as
(1)∇·v=0,
and
(2)∇·σ+ρf=0,
where v is the velocity vector, ρ is the density of the continuum, and f is the body force vector [49]. In the above, σ is the Cauchy stress tensor, which may be expressed as
(3)σ=τ−PI,
where P is the pressure, I is the identity tensor, and τ is part of the stress tensor associated with the viscosity [49]. In fiber suspensions, τ may be written as a function of the fiber orientation state within the flow, such that
(4)$$\tau =2\eta D+2\eta N_{p}A:D,$$
where η is the viscosity of the Newtonian fluid, and $N_{p}$ is the particle number that characterizes the intrinsic anisotropic effect of fibers on the flow rheology [50]. In Equation (4), ***D*** is the second-order rate of deformation tensor and A is the fourth-order fiber orientation tensor which is described in detail below. We note that a non-Newtonian flow model can better represent the rheology of polymer composite melts (e.g., shear thinning). However, incorporating a shear-dependent viscosity models (e.g., power law model, Carreau law model, etc.) increases the nonlinearity of the fully coupled problem considered in this study, leading to additional convergence issues beyond those discussed below. Therefore, the flow/fiber-orientation coupling is modeled as a highly viscous Newtonian flow in this work as in prior PDAM research (see e.g., [33,37,38]), leaving the use of a non-Newtonian fluid for future work.

Jeffery’s work [51] which describes the motion of a single rigid ellipsoidal particle in a pure shear flow forms the basis for most fiber orientation studies. More recently, Folgar and Tucker [48] extended Jeffery’s theory to analyze the interaction between fibers in a non-dilute fiber suspension using a fiber orientation distribution function. Further, Advani and Tucker proposed the fiber orientation tensor approach to quantify the fiber alignment state for concentrated suspension systems [46], which requires fewer independent variables than that of the Folgar–Tucker model [48]. The Advani–Tucker orientation tensor evolution equation with Folgar–Tucker orientation diffusion is written as
(5)$$\frac{DA}{Dt}=\left(A\cdot W-W\cdot A\right)+\lambda \left(D\cdot A+A\cdot D-2A:D\right)+2\;C_{I}\dot{\gamma }\left(I-3A\right),$$
where the second- and fourth-order fiber orientation tensors are defined respectively as
(6)A=〈pp〉 and A=〈pppp〉,

Here, p is the unit vector defining the orientation of a single rigid fiber [46]. The angle bracket “< >” in Equation (6) denotes orientation average over all directions, weighted by the probability distribution function of the orientation [50]. The fourth order orientation tensor is approximated through the widely applied orthotropic fitted closure [47]. We note that the effect of applying different closure approximations in our simulation would be an interesting future work. In addition, Favaloro and Tucker [52] summarized the fiber orientation kinetic equations including the more recently developed RSC and ARD models. Nevertheless, applying these orientation models brings in other parameters in the simulation, which lies beyond the major scope of this study and may be addressed in separate in-depth work.

The tensors W and D are the vorticity tensor and rate-of-deformation tensor (as introduced above) of the suspension flow, respectively, which can be written as
(7)$$D=\left(\nabla v+\nabla v^{T}\right)/2\;\mathrm{and}\;W=\left(\nabla v-\nabla v^{T}\right)/2,$$
where ∇v is the velocity gradient and the superscript T refers to the tensor transpose [49]. In addition, the parameter λ in Equation (5) is a function of fiber geometry. For an ellipsoidal fiber, λ can be evaluated as
(8)$$\lambda =\left[{\left(a_{r}\right)}^{2}-1]/[{\left(a_{r}\right)}^{2}+1\right],$$
where $a_{r}\left(g_{r}\right)$ is the hydrodynamic aspect ratio of the ellipsoidal fiber, and $g_{r}$ is the geometrical aspect ratio of the fiber (i.e., the ratio of the fiber length to diameter) [53]. Note that $a_{r}=g_{r}$ for ellipsoidal fibers which is not the case for fibers of other shapes (cf. Zhang et al. [53]). The last term appearing in Equation (5) is referred as the isotropic rotary diffusion that was first proposed by Folgar and Tucker [48], which includes the empirically obtained fiber interaction coefficient $C_{I}$, to provides a means for incorporating the effect of fiber–fiber interaction. Bay [54] proposed $C_{I}$ is a function of volume fraction $v_{f}$ and hydrodynamic aspect ratio as
(9)$$C_{I}=0.0184\exp\left(-0.7148v_{f}a_{r}\right),$$

The dimensionless particle number $N_{p}$ in Equation (4) is of great importance as it defines the amount of anisotropy of the melt viscosity. Several models have been proposed to address material anisotropy for fiber suspensions for different fiber concentrations (see e.g., [55,56,57]). Herein, we adopt the rheological constitutive equation presented by Dinh and Armstrong [56] due to its simple implementation and relatively high reliability [44]. The Dinh–Armstrong model describes the effect of fibers on viscosity in terms of fiber orientation parameters including fiber aspect ratio and fiber volume fraction. For narrow-gap shear flows, Dinh and Armstrong defined $N_{p}$ as
(10)$$N_{p}=Hv_{f}/\left(1+Sv_{f}\right),$$
where $v_{f}$ is the fiber volume fraction, and H and S are material coefficients related to the fiber suspension system [44]. For suspensions with a fully aligned orientation state (which is often approached in PDAM nozzle flow, e.g., [32,33,34,35]), H is given by (cf. [58])
(11)$$H={\left(g_{r}\right)}^{2}/\;\left[3ln\left(\sqrt{\pi /v_{f}}\right)\right],$$
and S=0, as suggested by the Dinh–Armstrong model assuming that the particle’s thickness can be ignored [58]. The model assumes fibers behave as slender-bodies, which allows for hydrodynamic interactions between fibers, at least in an average sense. In this approach, simulations can be performed beyond the dilute regime, while Jeffery’s solution for flow around a single ellipsoid are limited to the dilute case [58]. This is pivotal in the analyses the short-fiber polymer composite systems used in PDAM, where most filled polymers are highly concentrated fiber suspensions.

### 2.2. Finite Element Simulation

The fully coupled flow and fiber orientation problem has been solved in prior research (e.g., [41,44]). It is common to use the finite element method to compute the melt flow velocity and the fiber orientation tensor field is solved by integrating orientation evaluation equation (i.e., Equation (5)) along the flow streamlines (e.g., [33,34,35,41,42,43]). Alternatively, in this work, we use an approach as in Wang and Smith [44] (and also VerWeyst and Tucker [50]) who employed the finite element method directly to both the flow and fiber orientation governing equations. In this approach, we employ the standard Galerkin finite element approach to compute elemental nodal velocities $d^{e}$, such that
(12)$$K^{e}d^{e}=F^{e},$$
where the element stiffness matrix $K^{e}$ is
(13)$$K^{e}=\int_{\Omega }\left[{\left(B_{s}^{e}\right)}^{T}\;\tilde{V}\;B_{s}^{e}\right]d\Omega +\gamma_{e}\int_{\Omega }\left[{\left(B_{s}^{e}\right)}^{T}\;1\;1^{T}\;B_{s}^{e}\right]d\Omega ,$$
and the element nodal force vector $F^{e}$ is
(14)$$F^{e}=\int_{\Omega }\left[\;\rho \;{\left(N^{e}\right)}^{T}f\right]d\Omega +\int_{\Gamma_{\sigma }}\left[\;{\left(N^{e}\right)}^{T}\;\vec{t}\;\right]d\Gamma ,$$
where $B_{s}^{e}=\nabla_{s}N^{e}$, $N^{e}$ is the elemental interpolation function matrix, and$\;\nabla_{s}$ is symmetric gradient operator for a 2-D axisymmetric coordinate system (see. Reddy [59]). In the above, $\tilde{V}$ is an integrated form of viscosity matrix that simplifies the weak form of the momentum equation to the concise format shown here (see e.g., [44]).

Similarly, the finite element form of the fiber orientation governing equation can be written as
(15)$$K_{a}^{e}d_{a}^{e}=F_{a}^{e},$$
where the element fiber orientation system matrix $K_{a}^{e}$ is
(16)$$K_{a}^{e}=\int_{\Omega }\left[\frac{{\left(N_{a}^{e}\right)}^{T}N_{a}^{e}}{\Delta t}+v\cdot \nabla \left(N_{a}^{e}\right)\right]d\Omega ,$$
and the related element nodal ‘force’ vector $F_{a}^{e}$ is
(17)$$F_{a}^{e}=\int_{\Omega }\left[\frac{{\left(N_{a}^{e}\right)}^{T}N_{a}^{e}}{\Delta t}d_{a}^{e}{|}^{q-1}+m^{q}\right]d\Omega .$$
and $d_{a}^{e}$ is the elemental nodal velocity and second-order orientation tensor solutions, respectively [44]. Specifically, $d_{a}^{e}$ is a vector of five independent variables in the second order orientation tensor such that$\;d_{a}^{e}={\left[A_{11},A_{12},A_{13},A_{22},A_{23}\right]}^{T}$, and $N_{a}^{e}$ is the elemental interpolation function vector for $d_{a}^{e}$. Similarly, m contains the same five components $\frac{DA_{ij}}{Dt}$ corresponding to $A_{ij}$ components in Equation (5) [44]. Lastly, the element matrices and vectors appearing in Equation (15) are assembled to form the global finite element matrix equations in the usual manner.

The fiber orientation finite element Equation (15) is a nonlinear system, which we solve using the Newtonian–Raphson (N-R) iterative method [60] where the tangent matrix of Equation (15) is computed using the forward finite difference method with a step size of $10^{-5}$ for $d_{a}^{e}$, which yields trivial difference as compared to results computed by applying $10^{-6}$ as the step size. In addition, we note that the steady state fiber orientation solution within the flow domain is computed with an implicit time marching method (i.e., first order backward finite difference method) as in [44,50] and index q appearing in Equation (17) refers to the N-R iteration. We note that applying finite difference approximation in computing the tangent matrix of Equation (15) reduces the convergence rate from the quadratic form of regular N-R method, while the accuracy of the solved nodal solutions is retained as indicated in [50]. A detail derivation process for the above is omitted for conciseness and can be found in the dissertation of the first author [61], including the elemental stiffness matrices and force vectors, and the tangent matrix of the finite element system, etc.

The Galerkin finite element solution of the second-order fiber orientation tensor (cf. Equations (15)–(17)) may exhibit spatial instabilities due to the lack of a diffusion term in the governing equation (cf. Equation (5)) [44]. Herein, we apply the Streamline Upwind Petrov Galerkin (SUPG) method [62] to stabilize the nodal orientation tensor solution of $d_{a}^{e}$ in Equation (15). The SUPG method replaces the Galerkin weight function (i.e., $w=N_{a}^{e}$) with
(18)$$\tilde{w}\left(v,w\right)=N_{a}^{e}+\frac{\alpha h_{e}\left(v\right)}{2\left|\left|v\right|\right|}\nabla N_{a}^{e},$$
where α is a scale factor controlling the magnitude of the streamline upwinding [63]. Here, we set α=0.5 as in Wang and Smith for a similar problem [44] which provides just enough upwinding to eliminate spatial instabilities without degrading the orientation tensor solution.

### 2.3. Extrusion-Deposition Flow Modelling

This paper considers large area extrusion deposition additive manufacturing of short fiber polymer composites. The flow domain of interest includes the internal geometry of the extrusion die and a single layer of a planar deposited bead on a translating substrate as shown in Figure 2a. The extrusion-deposition process modeled here occurs in a short period of time justifying an isothermal assumption such that thermal effects are not included in the computation. Note that non-isothermal studies for similar flow geometries were performed using other simulation approaches (e.g., [42,43]). A 2D planar flow domain is employed where a unit width (i.e., the out-of-plane dimension) is assumed for the deposited bead (similar to that appearing in [38,41,42,43]). The nozzle internal flow geometry is based on the Strangpresse large-scale additive manufacturing Model 19 single screw extruder nozzle (cf. Figure 2b). The height of the deposited bead is 3 mm (i.e., distance between nozzle end and deposition substrate), where the outer surface of the nozzle is assumed to yield a negligible effect on the flow surface of the planar deposited bead [38]. The ratio of the deposited bead length to its height is 10:1, which ensures a steady-state condition of the flow and fiber orientation is achieved within the modeled flow domain. It should be noted that the relative distance between the nozzle exit and the substrate is of great importance in form the bead height, however, this factor is out of the main scope of this research and are discussed in prior works (see e.g., [38,42]).

The planar deposition flow model appears in Figure 3, where the flow domain is separated into three subdomains including the internal nozzle flow domain $\Omega_{1}$, the deposition turning flow domain $\Omega_{2}$, and the deposited bead domain $\Omega_{3}$. The boundary conditions for the flow domain in Figure 3 are as follows:

$\Gamma_{1}$: Flow domain inlet referring to the extrusion nozzle inlet, where the prescribed volumetric flow rate Q = 304.8 ${\mathrm{mm}}^{3}/s$ is specified (i.e., averaged velocity at the flow inlet is 24 mm/s).$\Gamma_{2}$: No slip wall boundary assuming no polymer sticking to the internal surface of the nozzle, where $v_{s}=v_{n}=0$.$\Gamma_{3}$: Free surface boundary simulating the free out-surface of the deposited bead, where v·n=0. Notice, this boundary condition is imposed multiple boundaries of subdomains $\Omega_{2}$ and $\Omega_{3}$.$\Gamma_{4}$: Contact surface between deposited bead and substrate, where $v_{s}=101.6\;\mathrm{mm}/s$ is the velocity of the moving deposition platform, $v_{n}=0$.$\Gamma_{5}$: Flow domain exit referring to a place that quasi-steady fiber orientation state of a deposited bead could be achieved, where $v_{n}=101.6\;\mathrm{mm}/s$, $v_{s}=0$.

In the above, $f_{s}$ is the tangential traction, $f_{n}$ is the normal traction, $v_{s}$ is the tangential velocity, and $v_{n}$ is the normal velocity. Note that the prescribed volumetric flow rate Q and the normal velocity of boundaries $\Gamma_{4}$ and $\Gamma_{5}$ are specified to simulate the large scale deposition flow rate of the Model 19 extruder, as in [38]. Further, v is the velocity vector at the free surface, and n is the free surface unit normal vector. A zero normal velocity boundary condition on the free surface is imposed by implementing the 1-D streamline-wise remeshing technique proposed by Tanner et al. [45], which assumes that the free surface boundary of the planar extrudate forms a streamline along the boundary of the flow domain. The direction of the polymer composite melt flow goes from a vertical to a horizontal flow in the deposition turning flow domain $\Omega_{2},$ making it helpful to divide the deposited bead into two subdomains $\Omega_{2}$ and $\Omega_{3}$ to simulate the free surface boundaries before and after the flow contacts the substrate, respectively, as shown in Figure 4a. The points $P_{0}^{\Omega_{2}}$ and $P_{0}^{\Omega_{3}}$ denotes the nodes at the intersection of the fixed-wall and free surface boundaries. To enforce the zero normal velocity on the free boundary nodes, we use the relation
(19)$$\delta_{i+1}^{\left(j\right)}=\delta_{i}^{\left(j\right)}+\int_{\omega_{i}}^{\omega_{i+1}}\frac{v_{n}}{v_{s}}d\omega ,$$
where ω and δ refer to the coordinates in the direction of flow and perpendicular to the direction of the flow, respectively. The subscript i indicates successive node numbers which start at 0 with the fixed node at the end of the no-slip wall [45], e.g., $P_{0}^{\Omega_{2}}$, $P_{1}^{\Omega_{2}}$, $P_{2}^{\Omega_{2}}$ appearing in Figure 4b. From above, it is seen that the ω coordinates of nodes in $\Omega_{2}$ do not vary. Note, ω refers to different directions in $\Omega_{2}$ and $\Omega_{3}$, due to the 90 degree turn of the deposition flow. The free surface is computed in an iterative manner where we convergence is assessed with the die swell ratio of the extrudate, which is written as
(20)$$B^{\left(j\right)}=\delta_{k}^{\left(j\right)}/\delta_{0},$$
where B donates the die swell ratio, $\delta_{k}^{\left(j\right)}$ refers to the normal-to-flow-direction coordinate of the k-th node of the free surface along the flow direction (i.e., typically the last node of the surface boundary which indicate a steady state) at the j-th iteration. The convergence of the flow surface is achieved when the error between the steady state die swell ratio values in two successive iterations $B^{\left(j\right)}-B^{\left(j-1\right)}\leq 10^{-2}$, from which the computed result shows good agreement with the free surface shape computed with a commercial finite element solver for the uncoupled flow problem as shown in the Appendix A. The δ coordinates of internal nodes are updated by a linear interpolation between those of the free surface and the fixed line of the subdomain (i.e., the intersection line between $\Omega_{2}$ and $\Omega_{3}$ is the fixed line for $\Omega_{2}$, and the contact surface between the deposition flow and the substrate serves as the fixed line for $\Omega_{3}$). Additionally, the initial mesh of the flow domain is generated using ANSYS Polyflow module assuming a domain boundary without die swell. The mesh of the flow domain remains unchanged except that affected by the change in location of the free surface boundaries. The overall flow computation continues until both free surface shapes meet the convergence criterion described above.

## 3. Results and Discussions

In this study, we use the 13 wt.% CF/ABS as the polymer composite melt material model that has seen widespread applications in the large area polymer deposition additive manufacturing [2,27,36]. We assume the Newtonian viscosity of the melt is η=817 Pa·s, which is the shear viscosity of the ABS polymer at 230 °C and shear rate 100 $s^{-1}$, and the fluid density is ρ = 1154 kg/m^3^, both of which are from Heller et al. [38]. In addition, we assume an isotropic fiber orientation field for the entire flow domain as a starting point for the fiber orientation calculation iterations when computing the fully coupled flow-orientation solutions, as in [44,50]. Moreover, the hyperbolic form of Equation (5) requires an initial condition of the second-order fiber orientation tensor at the flow inlet. Herein, we imposed the fiber orientation state of a fully developed flow, which assumes that the orientation state reaches steady state upstream of the flow inlet. The initial condition imposed at the flow inlet is fixed as $A^{ini}$, such that
(21)$$A^{ini}=\left[\begin{array}{ccc} 0.06744 & 0.00036 & 0\\ 0.00036 & 0.77956 & 0\\ 0 & 0 & 0.153 \end{array}\right],$$
which is computed using the uncoupled flow kinematics of a long tube flow (in the same diameter as the nozzle die inlet). Moreover, to reduce the singularity issue along the flow boundaries, fiber orientation along the no-slip wall is specified as fully aligned in the extrusion direction, and fiber–wall interaction is neglected as suggested by [64]. This fiber alignment assumption enhances the convergence behavior of the iterative process for the fiber orientation tensor field, while having little effect on the accuracy of velocity and fiber orientation solution [44]. We note that the boundary conditions specified above for the fiber orientation calculation are fixed throughout the iterative process.

The fully developed velocity profile imposed at the flow inlet (cf. Figure 3) is computed in a separate analysis using the ANSYS Polyflow module based on the prescribed flow rate Q and the selected rheology model, and is fixed throughout the flow computation process. The flow domain in Figure 3 is meshed with 4-node quadrilateral elements using ANSYS. There are a total of 3550 elements with 3736 nodes in our flow model, which was found to provide a sufficient and efficient mesh quality through our mesh sensitivity study. Additionally, the fully coupled problem is solved in a decoupled fashion, where the flow and fiber orientation finite element matrices are computed independently with the counterpart fixed. The two sub-solutions are altered iteratively until both solutions converge, such that
(22)∥deh−deh−1∥/∥≤10−2 and, ∥daeh−daeh−1∥/∥daeh∥≤10−2,
where “∥ ∥” refers to the Frobenius vector norm [60], and *h* is the iteration index of the overall coupling scheme. The convergence criterion appearing in Equation (22) indicates the absolute relative difference of two successive solutions are less than 1%, which yields sufficient accurate results as shown below.

By comparing results between uncoupled and coupled solutions, mutually dependent effects between the flow and fiber orientation in the planar deposition flow can be clearly illustrated. Below, we consider the following computed outputs for comparison: flow kinematics throughout the flow fields, orientation tensor fields within the flow domain, and the mechanical properties at the flow end $\Gamma_{5}$ that also represents the mechanical performances of deposited beads. In addition, a parametric study is performed on $C_{I}$ to assess the sensitivity of the computed coupling effects with respect to the fiber–fiber interaction coefficient.

### 3.1. Flow Kinematics

Results from the uncoupled flow kinematics simulated with $N_{p}$ = 0 computed using our custom melt flow simulation presented above show good agreement with those obtained from the commercial finite element suite ANSYS Polyflow. For conciseness and completeness, the data comparison is given in the Appendix A. In the following, we discuss the results of the extrusion/deposition flow and planar extrudate swell computed using both our uncoupled and our fully coupled computational methods.

Contours of computed melt flow velocity $v_{x}$ and $v_{y}$ appear in Figure 5 and Figure 6, respectively. Upon comparing the uncoupled and fully coupled results, significant variations can be seen in both $v_{x}$ and $v_{y}$ as the melt flow passes through the Nozzle Convergence Zone (NZR) and Deposition Transition Zone (DTZ). Further, contours of the velocity differences $v_{x}^{c}-v_{x}^{unc}$ and $v_{y}^{c}-v_{y}^{unc}$ appear in Figure 7 and Figure 8, respectively, where the NCZ and DTZ are separately specialized (note superscripts c = coupled and unc = uncoupled). It can be seen that the maximum difference in flow contours appears in the DTZ of $v_{x}$ (cf. Figure 7b), where the fully coupled velocity is about 14 mm/s higher than that of the uncoupled solution. This velocity difference is significant since it is 13.8% of the deposition rate of the flow (i.e., 101.6 mm/s, as imposed at the flow end). The fiber orientation coupling effects also induce a notable change in $v_{y}$ near the boundaries of the convergence zone, while the center of the melt flow slows over 10 mm/s (as indicated in Figure 8a). Moreover, the polymer melt die swell is also affected by the coupling. The shape of the free surfaces of deposited beads appearing in Figure 9 show that the maximum die swell ratios (cf. Equation (20)) of the bead front swell profile are 1.31 and 1.24 as computed with the uncoupled and coupled formulation, respectively. That is, the presence of fibers reduces the bead front swell by ~6% as compared to that of virgin ABS polymer deposited bead. In addition, the free surface profile of the turning flow (i.e., $\Omega_{3}$ in Figure 3) predicted by the fully coupled formulation increases slightly as compared to that of the uncoupled solution and the die swell ratios at the flow end solved by the uncoupled and fully coupled formulations are 0.95 and 0.96, respectively. The computed results shown above indicate that processing parameters such as the extrusion flow rate, deposition rate as well as the inter-beads spacing for virgin polymer and their filled composite counterpart should be different, and proper adjustment are needed to compensate for the differences yielded by the fully coupled interactions between the flow kinematics and the fiber orientations.

### 3.2. Second-Order Fiber Orientation Tensor Fields

Our fully coupled algorithm is first tested with the same parameter inputs as in Heller et al. [38]. A fixed time step is adopted in time marching scheme in order to obtain the steady state fiber orientation state in the flow starting from an isotropic fiber alignment (i.e., fully random fiber orientation state). Herein, the time increment is fixed at 0.01 which was found to converge well and to be computationally efficient. Convergence is assumed when the absolute relative difference (cf. Equation (22)) of orientation tensor solutions between two successive time steps is less than $5\times 10^{-3}$. We note that convergence issues were experienced when imposing a tighter convergence criterion while the fiber orientation tensor solution at the flow end with current convergence criterion show a good agreement with those explicitly given in Heller et al. [38] (cf. Figure 10, where “myfea” refers to results obtained by our finite-element-based algorithm). Therefore, it can be expected that the adopted convergence criterion for the fiber orientation solution yields reliable numerical results and thus is applied in the following studies.

For the material model of this study, we consider a CF/ABS composite having a weight fraction of 13% (i.e., the volume fraction of carbon fiber is roughly $v_{f}=$ 8.4% [35]), a constant fiber aspect ratio $g_{r}=$30, as defined in related literature (i.e., mean fiber length is 150 μm [65], and a mean fiber diameter of 5 μm [27]). Consequently, λ, $C_{I}$, and $N_{p}$ are determined as 0.9950, 0.0055, and 13.92, respectively, using Equations (8)–(11). Computed contours of second order orientation tensor components $A_{22}$, $A_{12}$, and $A_{11}$ are given in Figure 11, Figure 12 and Figure 13, respectively. Note that the fiber orientation tensor components $A_{22}$ and $A_{12}$ change significantly as the flow passed through the DTZ. To make the variations within a certain flow region more obvious, the color scale of the presented contours is adjusted. It is seen that the fibers are highly aligned along the direction of the flow downstream of the nozzle convergence zone. Specifically, the vortex in the nozzle convergence corner grows larger in the fully coupled simulation as compared to the uncoupled solution. In addition, there is a notable difference between the uncoupled and fully coupled solution near the wall boundaries of the NCZ in both the results of $A_{22}$ and $A_{12}$ (cf. Figure 11 and Figure 12), which may be a result of the pre-defined wall-alignment fiber orientation boundary condition (cf. [64]).

The fiber orientation formation in the DTZ is of special interest. To this end, we plot the strain rate contours (i.e., velocity gradients) in Figure 14, Figure 15, Figure 16 and Figure 17 to better explain the fiber alignment variations that occur during the transition between extrusion and planar deposition. It is clearly seen that the fiber alignment changes rapidly in the DTZ, which is a result of the extensional flow forming during the transition from extrusion to deposition (cf. Figure 14a and Figure 17a, where significant variations appear in the DTZ). Figure 13 indicates that the fully coupled solution and the uncoupled solutions of $A_{11}$ exhibit a similar orientation pattern as the deposition flow reaching a quasi-steady-state, except that the upper region in the DTZ exhibits an increment of ~0.1 in the fully coupled solution (cf. Figure 13c). This is expected to result from the non-uniform shear force that is applied to the melt as the extrudate contacts the substrate. This change is captured by the fully coupled simulation (cf. the strain rate variation along the y-direction, i.e., Figure 15 and Figure 17, where large increments appear in the contacting region). Fibers near the free surface boundary of the deposition flow (i.e., free surface of $\Omega_{3}$) recover to high alignment along flow direction, which is likely due to that the deposition flow evolves into a shear dominate flow as shown in Figure 15). Similar results are also reported in Ouyang et al. [43]. Specifically, variations between the uncoupled and coupled solutions can be seen near the DTZ (cf. Figure 13c), where the fully coupled solution exhibits a reduced fiber alignment in the front of the deposition flow and the values of $A_{11}$ increase notably in the middle of the DTZ as the flow is redirected towards front of the deposition. The noticeable differences between the uncoupled and fully coupled fiber orientation solutions within the extrusion-deposition flow domain imply the importance of the fully coupled flow/fiber-orientation simulation in interpreting the process-structure-properties mapping of PDAM applications of composite materials, as suggested in [66].

### 3.3. Fiber–Wall Interaction in Nozzle Convergence Zone

Rosenberg and Denn [67] (also see VerWeyst and Tucker [50]) stated that fiber–wall interaction at the no-slip wall boundary of a flow suspension causes a singularity where the finite length fibers exhibit a periodic tumbling motion within the flow and the fiber orientation tensor yields (e.g., Equation (5)) a unique pattern at the wall boundary. To reduce the intensity of this singularity, we set the fiber orientation to align in the direction of the flow at the wall boundaries as suggested by Ranganathan and Advani [64]. Unfortunately, the flow contraction in the nozzle convergence zone exhibited numerical instabilities in the fiber orientation tensor solution, especially in the fully coupled simulation (cf. Figure 11b and Figure 12b). Figure 18 shows a plot of fiber orientation from the uncoupled and fully coupled solutions near the nozzle convergence zone, where the length and direction of each line segment is determined by the largest eigenvalue of the second order orientation tensor and its associated eigenvector, respectively [50]. In the uncoupled solution, fibers near the NCZ boundary are mis-aligned, especially in the vicinity of the nozzle exit. While in the case of the fully coupled simulation, fiber exhibit a higher misalignment where the flow wall boundary above the NCZ also shows such effect in addition to that of the NCZ. These results indicate that the flow–fiber coupling effect promotes the vortex generation in contraction region (i.e., corner of a converging flow as indicated by the quad block appearing in Figure 14). It is noted that a similar flow geometry (i.e., 4.5:1 contraction flow without nozzle convergence zone) was studied in Lipstomb et al. [68] and VerWeyst and Tucker [50] where the corner vortex exhibited a higher increment when the fully coupled effect was considered, which compares well to our result in Figure 18. This implies that flow in the nozzle convergence zone may be a factor in reducing the corner vortex phenomenon of the fiber suspension, especially when highly concentrated composites are employed. We expect that the relaxed convergence criterion adopted for the fiber orientation iterative solution (i.e., error of orientation tensor solutions between two successive time steps reduces below $5\times 10^{-3}$) likely results in the numerical instability in the computed solution near the nozzle convergence zone boundaries (see e.g., Figure 11b). In addition to enhance the convergence behavior of the simulation, we also check the converged solution at each time step, where the non-physical orientation tensor eigenvalues (i.e., diagonal terms of the tensor is above one) are adjusted to be within the physical bounds (i.e., within an interval from zero to one) and the associated eigenvectors are modified as well as in [50].

### 3.4. Sensitivity on CI Parameter

The fiber–fiber interaction coefficient $C_{I}$ in Equation (5) has a significant influence on the fiber orientation prediction, especially in fully coupled flow–fiber simulations [50]. Equation (9) proposed by Bay [54], which we use to compute $C_{I}$, empirically determined from injection molding studies, has not been fully validated for polymer deposition. To explore the sensitivity of the fully coupled simulation results to the fiber interaction coefficient, we consider various values of $C_{I}$ in the interval from 0.001 to 0.01, which is typical region for short fiber polymer composites studies [33,34,35,36,37,38]. Computed orientation tensor component values for $0.001\leq C_{I}\leq 0.01$ at the nozzle exit and the flow end appear in Figure 19, Figure 20, Figure 21, Figure 22, Figure 23 and Figure 24, respectively, in which the standard deviations among the numerical data are evaluated as well. In all cases, computed results show that fibers are highly aligned along the flow direction. Figure 21 (i.e., $A_{22}$ refers to flow-direction alignment) and Figure 22 (i.e., $A_{11}$ refers to flow-direction alignment) show that $C_{I}$ = 0.001 yields the highest flow-direction alignment among all simulations, which implies that an increased fiber–fiber interaction reduces fiber rotation that aligns fibers the direction of flow. While the in-nozzle fiber orientation is symmetric with respect to the central axis of the nozzle, it is clearly seen that the orientation state across the bead is not, and instead changes as the melt comes into contact with the substrate. As a result, flow-direction alignment is higher in the lower region of the deposition flow (i.e., subdomain $\Omega_{3}$) than along the upper surface as seen in Figure 22, Figure 23 and Figure 24. This pattern of fiber alignment is due in part to the shear force applied by the moving substrate. By reducing the value of $C_{I}$, the flow-direction fiber alignment increases which can be as high as that near the substrate-contact boundary for $C_{I}$ = 0.001. In addition, the standard deviations appearing in the Figure 19, Figure 20, Figure 21, Figure 22, Figure 23 and Figure 24 indicate that the employed value of $C_{I}$ mostly affects the flow-direction alignment, as evidenced by the deviation of ~0.07 for $A_{22}$ at nozzle exit and ~0.10 for $A_{11}$ at deposition flow end.

Finally, the fiber orientation state in the polymer melt flow is a pivotal factor in determining the material properties of the deposited beads. To assess the effect of processing on mechanical properties, we apply the orientation-homogenization approach [46] to compute the material stiffness and thermal expansion coefficient of the composite bead using second order fiber orientation tensor values computed at the flow end (e.g., computed results appearing in Figure 22, Figure 23 and Figure 24). The flow end is chosen here since this location represents the steady-state orientation pattern of a deposited bead. Properties of the constituent materials used in this study are given in Table 1. Furthermore, the computed properties at the nodal positions are numerically integrated along the y-axis over the end of the bead in order to obtain the effective mechanical property of interest. More detail on material property prediction method used here can be found in [33]. Computed results are given in Table 2, Table 3 and Table 4, where the effective axial and shear moduli are computed based on local predicted material stiffness values. The Partial Relative Differences (PRD) among computed data are considered here, where the properties evaluated by employing $C_{I}$ = 0.001 is designated as the reference data. The PRD may be evaluated as [37]
(23)$$PRD=\left|P_{1}-P_{2}\right|/\left(\left|P_{1}+P_{2}\right|/2\right)\times 100\%.$$

The resulting PRDs are given in Table 5, Table 6 and Table 7, where the differences yielded by varying $C_{I}$ values clearly shown. It can be seen that the elastic constant ${\bar{E}}_{11}$ is most influenced by the value of $C_{I}$, since ${\bar{E}}_{11}$ is the effective axial modulus along the bead which is also the major fiber alignment direction. Specifically, the PRD between ${\bar{E}}_{11}$ computed with $C_{I}$ = 0.01 and that computed with $C_{I}$ = 0.001 is 23.16%. It is also seen that the impact of $C_{I}$ choice is larger for elastic constants than the thermal expansion coefficient, where the PRDs for moduli (cf. Table 5 and Table 6) are ~10–20%, and those in $\bar{\alpha }$ are less than 10% (cf. Table 7). In addition, the transverse properties exhibit some skewness, especially for the computed thermal expansion coefficients. For example, in the case of $C_{I}$ = 0.01, ${\bar{\alpha }}_{22}$ is 13% higher than ${\bar{\alpha }}_{33}$. Uneven thermal expansions transverse to the bead axis may be a critical factor affecting the inter-bead void formation and distortion during manufacturing.

Additionally, we note that our computed values of effective modulus ${\bar{E}}_{11}$ show good agreement with experimentally reported values on the same material system with slightly different processing parameters (i.e., measured tensile modulus parallel to the printing direction of a 13 wt.% CF/ABS bead is reported as 8.18 GPa [27]). In addition, the predicted y-axis thermal expansion coefficient (i.e., ${\bar{\alpha }}_{22}$) properties also agree well with reported data for the same material system in a related study (i.e., 7.41 × 10^−5^/°C for FEA results and 5.77 × 10^−5^/°C for DIC measurement, see Table 1. in [31]). Herein, we consider the above favorable agreements as a support to the computed results from our fully coupled flow–fiber simulation. Nevertheless, we note that the employed fiber orientation constitutive equations (i.e., Equations (5)–(9)) describe the fiber–fiber interaction through a phenomenological manner, which we believe provides valuable insight into the flow of fibers in the large area additive manufacturing polymer composite deposition process. More advanced physical-based approaches have yet to be developed. It is important to note that, beyond our simulation, the microstructural voids formation [69], intense fiber–fiber interaction in highly concentrated fiber composites [65], and reduced fiber length distribution [36] can also significantly affect the material properties of polymer deposition additive manufactured composites.

## 4. Summary

This paper numerically characterizes the mutually dependent effects between polymer melt kinematics and suspended fiber orientation in polymer deposition additive manufacturing of short fiber reinforced composites. The numerical solution of the governing equations for flow and fiber orientation problems are computed iteratively using the finite element method. Calculated results indicate that the uncoupled and fully coupled flow field solutions exhibit significant differences near the nozzle convergence zone and the extrusion-deposition transition zone where intense flow/orientation mutual interaction occurs. The orientation tensor component $A_{22}$ exhibits notable variation in between the uncoupled and fully coupled solution along the nozzle convergence zone boundaries, where the fully coupled results experience significant numerical instability in the flow contraction region. The pattern of the $A_{11}$ component computed from both simulations are similar, where the largest difference is seen at the extrusion-deposition transition zone. The sensitivity study on $C_{I}$ indicates that fiber–fiber interaction is of great importance in determining the amount of fiber alignment along the flow domain, especially for the steady-state orientation at the flow end, which results in a 23% PRD error in ${\bar{E}}_{11}$ computed using $C_{I}$ = 0.01 and $C_{I}$ = 0.001. These results indicate that an in-depth numerical and experimental combined study on $C_{I}$ would be of great value for further explaining the fully coupled flow/orientation features in polymer deposition additive manufacturing applications. In addition, the transverse thermal expansion coefficients exhibit notable degree of skewness which may lead to uneven distortion during manufacturing. Lastly, our predicted material properties of 13 wt.% CF/ABS show favorable agreement with experiment data reported in the literature on the same material model which support the simulation approach and associated computed results.

## Figures and Tables

**Figure 1 materials-14-02596-f001:**
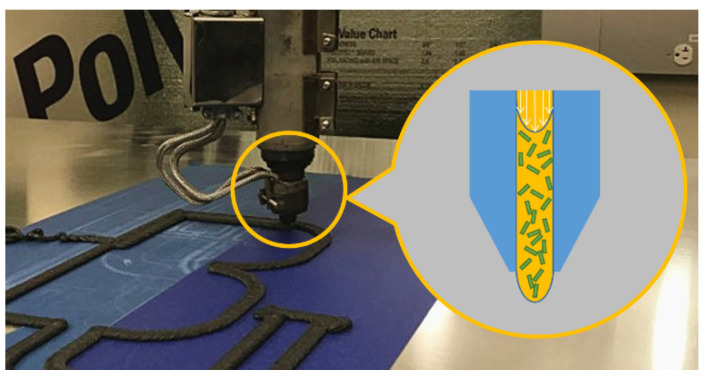
Manufacturing 13 wt.% CF/ABS with a Large-scale extruder-based PDAM system.

**Figure 2 materials-14-02596-f002:**
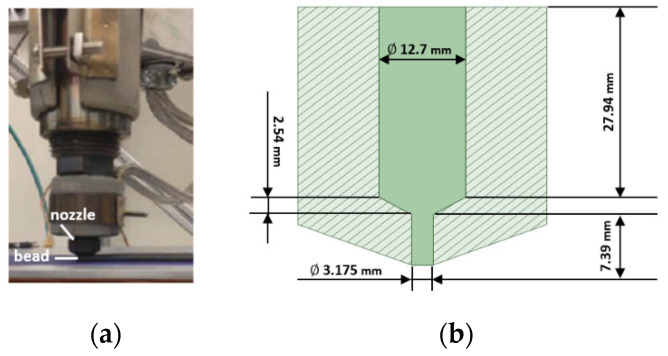
(**a**) Polymer composite deposition process; (**b**) dimensions of extrusion die mounted on the Strangpresse Model 19 extruder (solid shaded region is the flow domain inside the nozzle).

**Figure 3 materials-14-02596-f003:**
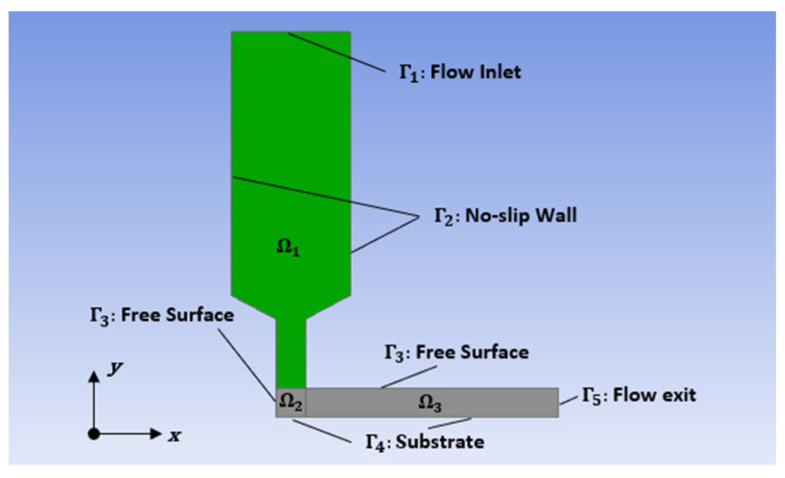
Planar deposition flow domain.

**Figure 4 materials-14-02596-f004:**
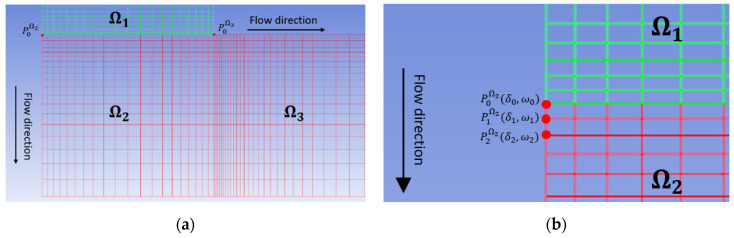
Free surface remeshing by streamline-wise method: (**a**) dividing the deposition flow into two subdomains; (**b**) example of applying the remeshing approach to free surface of $\Omega_{2}$.

**Figure 5 materials-14-02596-f005:**
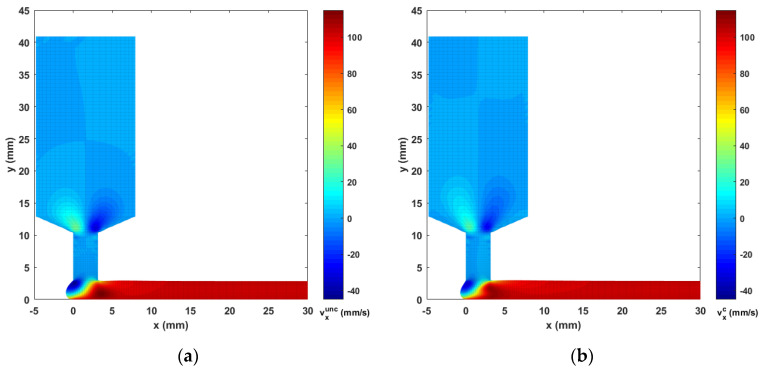
Velocity contours $v_{x}$ of (**a**) the uncoupled solution; (**b**) the fully coupled solution.

**Figure 6 materials-14-02596-f006:**
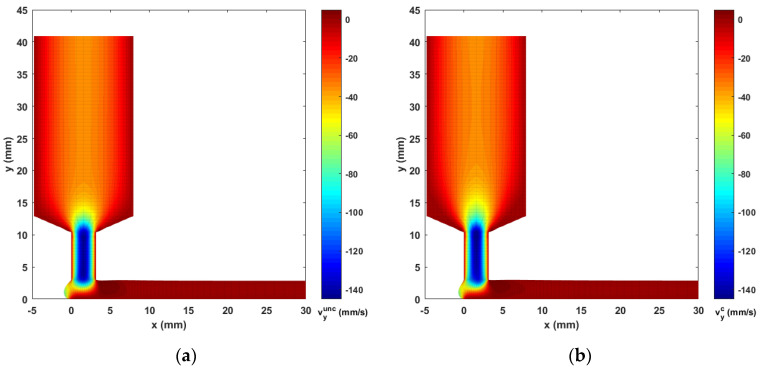
Velocity contours $v_{y}$ of (**a**) the uncoupled solution; (**b**) the fully coupled solution.

**Figure 7 materials-14-02596-f007:**
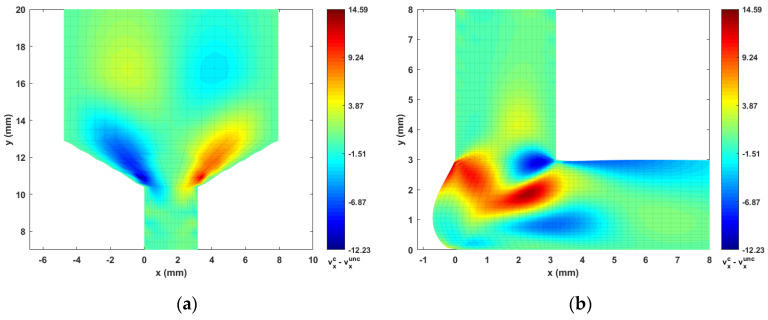
Differences of $v_{x}$ between the fully coupled solution and uncoupled solution: (**a**) NCZ; (**b**) DTZ.

**Figure 8 materials-14-02596-f008:**
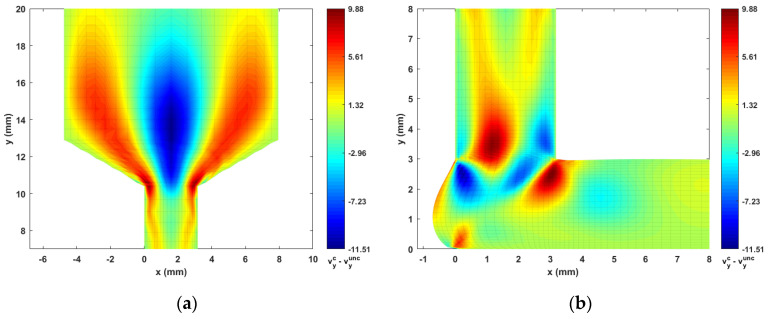
Differences of $v_{y}$ between the fully coupled solution and uncoupled solution: (**a**) NCZ; (**b**) DTZ.

**Figure 9 materials-14-02596-f009:**
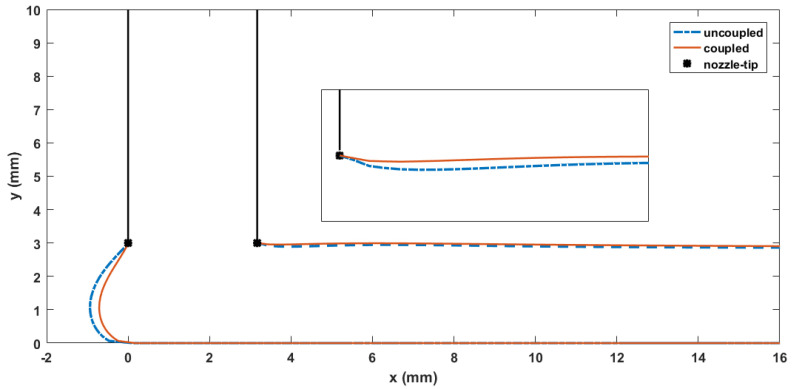
Free surface identifications of the planar deposition flow.

**Figure 10 materials-14-02596-f010:**
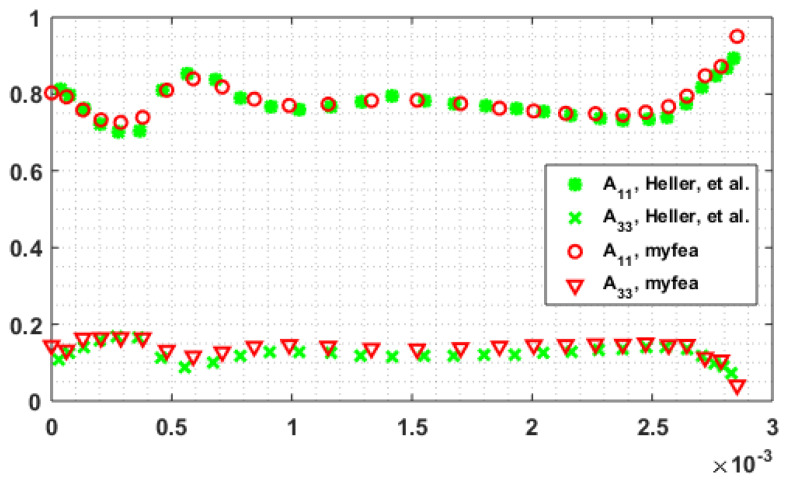
Fiber orientation diagonal components at the flow end.

**Figure 11 materials-14-02596-f011:**
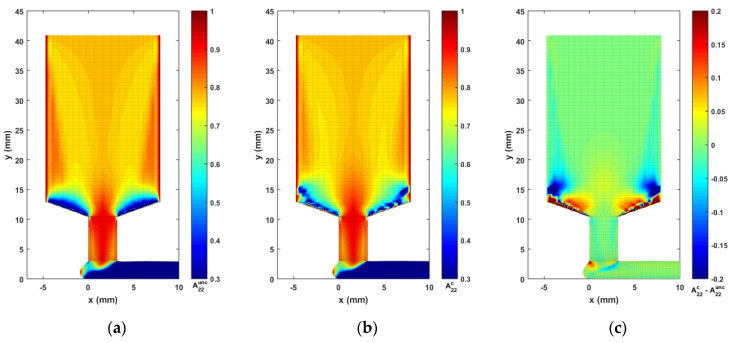
Fiber orientation tensor component contour of $A_{22}$: (**a**) the uncoupled solution; (**b**) the fully coupled solution; (**c**) difference between the fully coupled and uncoupled solutions.

**Figure 12 materials-14-02596-f012:**
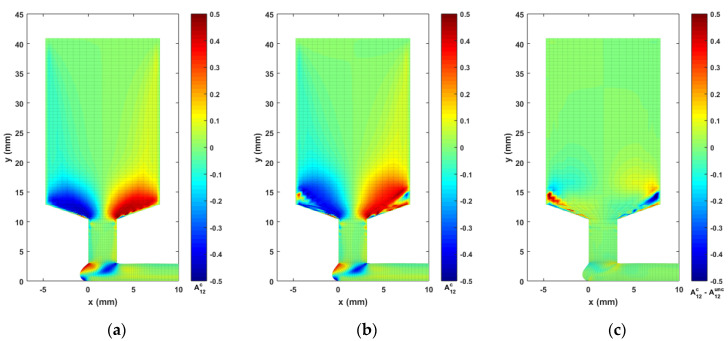
Fiber orientation tensor component contour of $A_{12}$: (**a**) the uncoupled solution; (**b**) the fully coupled solution; (**c**) difference between the fully coupled and uncoupled solutions.

**Figure 13 materials-14-02596-f013:**
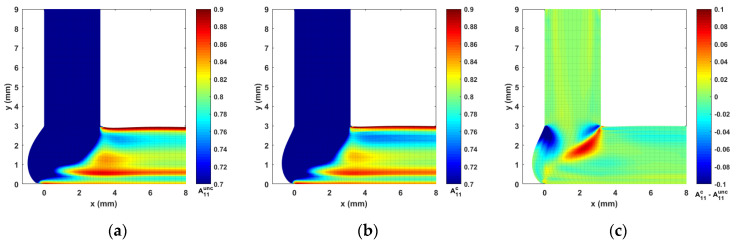
Fiber orientation tensor component contour of $A_{11}$: (**a**) the uncoupled solution; (**b**) the fully coupled solution; (**c**) difference between the fully coupled and uncoupled solutions.

**Figure 14 materials-14-02596-f014:**
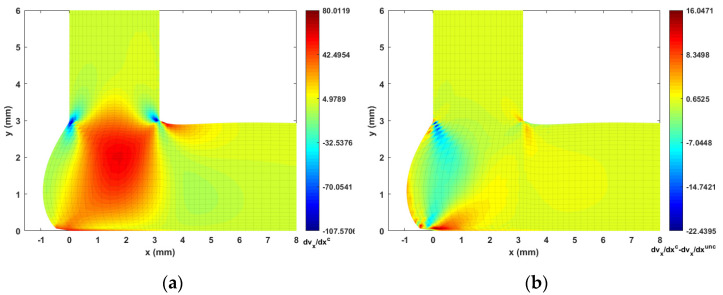
Velocity gradient contour $dv_{x}/dx$ near extrusion-deposition transition region: (**a**) fully coupled solution; (**b**) difference between the fully coupled and uncoupled solutions.

**Figure 15 materials-14-02596-f015:**
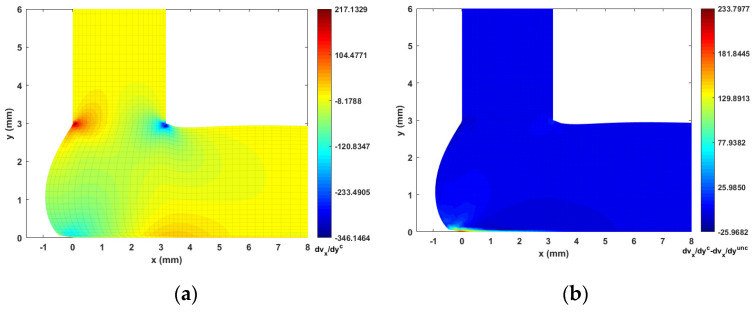
Velocity gradient contour $dv_{x}/dy$ near extrusion-deposition transition region: (**a**) fully coupled solution; (**b**) difference between the fully coupled and uncoupled solutions.

**Figure 16 materials-14-02596-f016:**
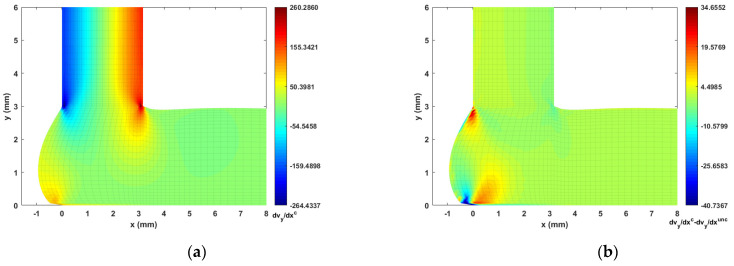
Velocity gradient contour $dv_{y}/dx$ near extrusion-deposition transition region: (**a**) fully coupled solution; (**b**) difference between the fully coupled and uncoupled solutions.

**Figure 17 materials-14-02596-f017:**
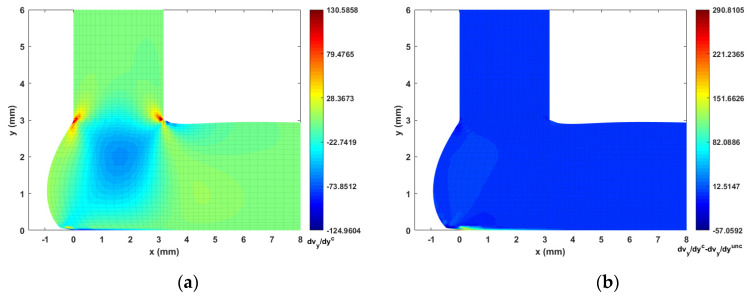
Velocity gradient contour $dv_{y}/dy$ near extrusion-deposition transition region: (**a**) fully coupled solution; (**b**) difference between the fully coupled and uncoupled solutions.

**Figure 18 materials-14-02596-f018:**
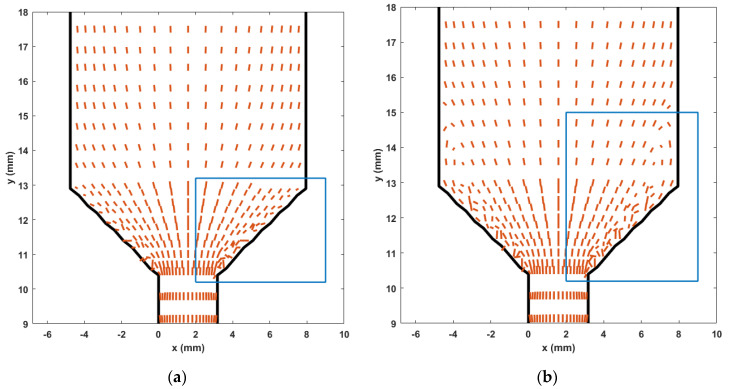
Fiber orientation tensor vector field: (**a**) uncoupled solution; (**b**) fully coupled solution.

**Figure 19 materials-14-02596-f019:**
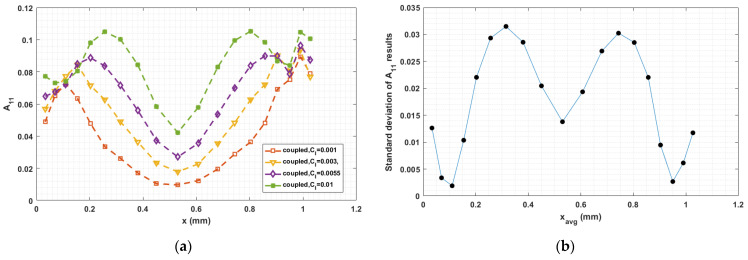
(**a**) Fiber orientation tensor component $A_{11}$ at nozzle exit; (**b**) standard deviation of the parametric study resulted $A_{11}$ versus the averaged x locations.

**Figure 20 materials-14-02596-f020:**
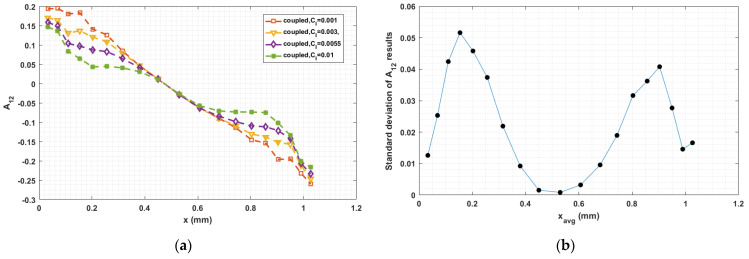
(**a**) Fiber orientation tensor component $A_{12}$ at nozzle exit; (**b**) standard deviation of the parametric study resulted $A_{11}$ versus the averaged x locations.

**Figure 21 materials-14-02596-f021:**
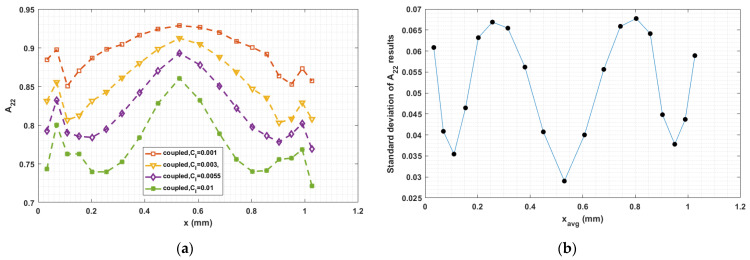
(**a**) Fiber orientation tensor component $A_{22}$ at nozzle exit; (**b**) standard deviation of the parametric study resulted $A_{11}$ versus the averaged x locations.

**Figure 22 materials-14-02596-f022:**
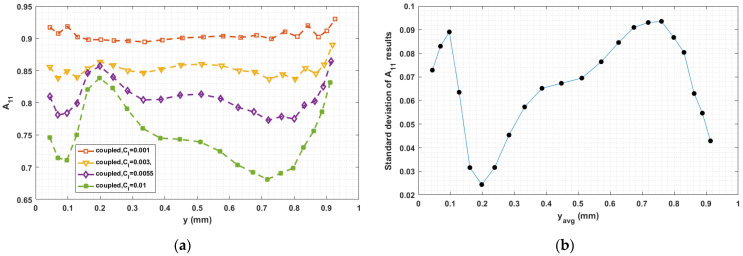
(**a**) Fiber orientation tensor component $A_{11}$ at flow end; (**b**) standard deviation of the parametric study resulted $A_{11}$ versus the averaged y locations.

**Figure 23 materials-14-02596-f023:**
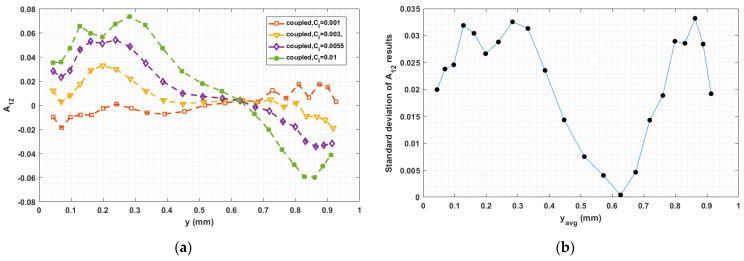
(**a**) Fiber orientation tensor component $A_{12}$ at flow end; (**b**) standard deviation of the parametric study resulted $A_{12}$ versus the averaged y locations.

**Figure 24 materials-14-02596-f024:**
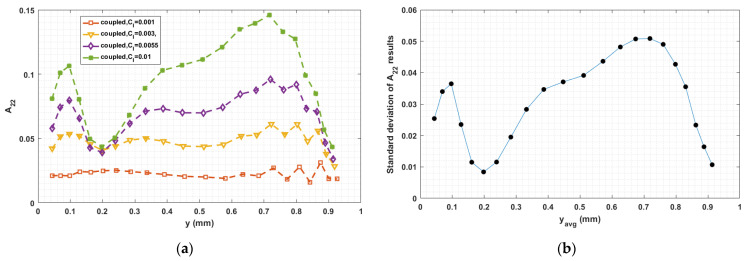
(**a**) Fiber orientation tensor component $A_{22}$ at flow end; (**b**) standard deviation of the parametric study resulted $A_{22}$ versus the averaged y locations.

**Table 1 materials-14-02596-t001:** Elastic constants of the constituents of a 13 wt.% CF/ABS [36].

Material	E (GPa)	ν	α (1/°C)
ABS	2.25	0.35	80 × 10^−6^
Carbon fiber	230	0.2	−0.6 × 10^−6^

**Table 2 materials-14-02596-t002:** Computed mean axial elastic constants of 13 wt.% CF/ABS deposited bead.

$C_{I}$	${\bar{E}}_{11}\;(\mathrm{GPa})$	${\bar{E}}_{22}\;(\mathrm{GPa})$	${\bar{E}}_{33}\;(\mathrm{GPa})$
0.001	8.43	2.78	2.88
0.003	7.84	2.82	2.94
0.0055	7.33	2.86	2.99
0.01	6.68	2.92	3.08

**Table 3 materials-14-02596-t003:** Computed mean shear elastic constants of 13 wt.%. CF/ABS deposited bead.

$C_{I}\;$	${\bar{G}}_{12}\;(\mathrm{GPa})$	${\bar{G}}_{23}\;(\mathrm{GPa})$	${\bar{G}}_{13}\;(\mathrm{GPa})$
0.001	1.01	0.94	1.17
0.003	1.09	0.98	1.24
0.0055	1.14	1.01	1.29
0.01	1..20	1.04	1.36

**Table 4 materials-14-02596-t004:** Computed mean coefficients of thermal expansion of 13 wt.%. CF/ABS deposited bead.

$C_{I}\;$	${\bar{\alpha }}_{11}$ (1/°C)	${\bar{\alpha }}_{22}$ (1/°C)	${\bar{\alpha }}_{33}$ (1/°C)
0.001	2.55 × 10^−5^	7.35 × 10^−5^	6.66 × 10^−5^
0.003	2.59 × 10^−5^	7.22 × 10^−5^	6.52 × 10^−5^
0.0055	2.65 × 10^−5^	7.11 × 10^−5^	6.39 × 10^−5^
0.01	2.77 × 10^−5^	6.96 × 10^−5^	6.18 × 10^−5^

**Table 5 materials-14-02596-t005:** PRDs of computed mean axial elastic constants of 13 wt.% CF/ABS deposited bead.

$C_{I}\;$	${\bar{E}}_{11}$	${\bar{E}}_{22}$	${\bar{E}}_{33}$
0.001	0	0	0
0.003	7.25%	1.43%	2.06%
0.0055	13.96%	2.84%	3.75%
0.01	23.16%	4.91%	6.71%

**Table 6 materials-14-02596-t006:** PRDs of computed mean shear elastic constants of 13 wt.%. CF/ABS deposited bead.

$C_{I}\;$	${\bar{G}}_{12}$	${\bar{G}}_{23}$	${\bar{G}}_{13}$
0.001	0	0	0
0.003	7.62%	4.17%	5.81%
0.0055	12.09%	7.18%	9.76%
0.01	17.19%	10.10%	15.02%

**Table 7 materials-14-02596-t007:** PRDs of computed mean coefficients of thermal expansion of 13 wt.%. CF/ABS deposited bead.

$C_{I}\;$	${\bar{\alpha }}_{11}$	${\bar{\alpha }}_{22}$	${\bar{\alpha }}_{33}$
0.001	0	0	0
0.003	1.72%	1.82%	2.02%
0.0055	3.92%	3.32%	4.13%
0.01	8.17%	5.50%	7.37%

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
