# Peer review of "A Fully Coupled Simulation of Planar Deposition Flow and Fiber Orientation in Polymer Composites Additive Manufacturing"

_materials, 2021, doi:10.3390/ma14102596_

Round 1
Reviewer 1 Report
In this contribution, the authors present finite-element simulations of fibre suspensions in a deposition flow relevant for additive manufacturing.
Parameters are chosen to model a particular carbon-filled butadiene composite.
The authors find a strong sensitivity of fibre orientations and resulting elastic modulus on the strength of fibre-fibre interaction coefficient.
Overall, I think this is an interesting study that deserves to be published in materials.
I have a few comments and observations that the authors might want to consider before publishing.
1. The constitutive equation employed here, Eq (5) with (9), was proposed in Refs [45,47] ad hoc.
Especially fibre interactions are modelled in a completely phenomenological manner.
Therefore, it is not clear how well this model works for the particular carbon-filled butadiene composite studied here, let alone whether the celebrated quantitative similarity to a single measured tensile modulus is meaningful.
Therefore, I strongly encourage the authors critically discuss their claims.
2. The statement at the end of page 5 - that the model includes hydrodynamic interactions - is misleading since Jeffrey's solution holds only for an isolated ellipsoid and hydrodynamic interactions between fibres are ignored.
3. The authors mention that they use the orthotropic closure and cite Ref [46]. There, however, the closure was used but no details are given. Probably better to cite Cintra and Tucker, J Rheol 1995.
In this context it is probably interesting to consider new closure approximations that have been suggested and tested more recently in Kroger, Ammar, Chinesta, J Non-Newton Fluid Mech 2008.
For these, the authors' statement - that more advanced models bring additional parameters - is not true.
4. Eq (7): typo in the definition of W, minus sign
5. Eq 19, 20: double subscript in Xn, Xs do not display well. Why not use a different symbol?
6. It might be worth comparing their implementation for a 4:1 contraction to the results presented in Ref [66].
Do the authors also observe the large change in the corner vortex compared to the Newtonian case?
7. A similar geometry was studied for polymer melt extrusion for 3D printing using different models in different parts of the flow by McIlroy and Olmsted, J Rheol 2017.
Author Response
Dear reviewer,
The authors greatly appreciate your valuable comments, which helps us improving the quality of our manuscript. Please find our attachment for the reply of your comments. For your convienience to track every change of the manuscript, the cover letter attached also included the reply to another reviewers.
Kind regards,
Zhaogui Wang

Reviewer 2 Report
There are some weaknesses through the manuscript which need improvement. Therefore, the submitted manuscript cannot be accepted for publication in this form, but it has a chance of acceptance after a major revision. My comments and suggestions are as follows:
1- Abstract gives information on the main feature of the performed study, but some details about the consider AM process must be added.
2- Authors must clarify necessity of the performed research. Objectives of the study must be clearly mentioned in introduction.
3- The literature study must be enriched. In this respect, authors must read and refer to the following papers: (a) https://doi.org/10.1016/j.prostr.2020.10.083 (b) https://doi.org/10.1016/j.jmsy.2021.02.018
4- Details of boundary conditions must be explained.
5- The main reference of each formula must be cited. Moreover, each parameters in equations must be introduced. Please double check this issue.
6- Standard deviation is the presented curves must be discussed. In addition, error in calculation must be considered and discussed.
7- In its language layer, the manuscript should be considered for English language editing. There are sentences which have to be rewritten.
8- The conclusion must be more than just a summary of the manuscript. List of references must be updated based on the proposed papers. Please provide all changes by red color in the revised version.
9-There are sentences which need reference. For example, the first sentence of the subsection 2.2.
10-As it has been mentioned in 2.3, isothermal assumption are considered. How the final results are valid and comparable to real conditions.
11-Authors should explain why fully-coupled solution exhibits a reduced fiber alignment.
12-Reason of the differences of fiber alignment in uncoupled and fully-coupled simulations.
Author Response
Dear reviewer,
The authors greatly appreciate your valuable comments, which helps us improving the quality of our manuscript. Please find our attachment for the reply of your comments (starts from Page 4). For your convienience to track every change of the manuscript, the cover letter attached also included the reply to other reviewers.
Kind regards,
Zhaogui Wang

Round 2
Reviewer 1 Report
I am happy with the authors' response to my comments.
Reviewer 2 Report
The paper has been improved and corresponding modifications have been conducted.